# Sparse Approximate Conic Hulls

**Gregory Van Buskirk, Benjamin Raichel, and Nicholas Ruozzi**
Department of Computer Science
University of Texas at Dallas
Richardson, TX 75080
{greg.vanbuskirk, benjamin.raichel, nicholas.ruozzi}@utdallas.edu

## Abstract

We consider the problem of computing a restricted nonnegative matrix factorization (NMF) of an $m \times n$ matrix $X$. Specifically, we seek a factorization $X \approx BC$, where the $k$ columns of $B$ are a subset of those from $X$ and $C \in \mathbb{R}^{k \times n}_{\geq 0}$. Equivalently, given the matrix $X$, consider the problem of finding a small subset, $S$, of the columns of $X$ such that the conic hull of $S$ $\varepsilon$-approximates the conic hull of the columns of $X$, i.e., the distance of every column of $X$ to the conic hull of the columns of $S$ should be at most an $\varepsilon$-fraction of the angular diameter of $X$. If $k$ is the size of the smallest $\varepsilon$-approximation, then we produce an $O(k/\varepsilon^{2/3})$ sized $O(\varepsilon^{1/3})$-approximation, yielding the first provable, polynomial time $\varepsilon$-approximation for this class of NMF problems, where also desirably the approximation is independent of $n$ and $m$. Furthermore, we prove an approximate conic Carathéodory theorem, a general sparsity result, that shows that any column of $X$ can be $\varepsilon$-approximated with an $O(1/\varepsilon^2)$ sparse combination from $S$. Our results are facilitated by a reduction to the problem of approximating convex hulls, and we prove that both the convex and conic hull variants are d-SUM-hard, resolving an open problem. Finally, we provide experimental results for the convex and conic algorithms on a variety of feature selection tasks.

## 1 Introduction

Matrix factorizations of all sorts (SVD, NMF, CU, etc.) are ubiquitous in machine learning and computer science. In general, given an $m \times n$ matrix $X$, the goal is to find a decomposition into a product of two matrices $B \in \mathbb{R}^{m \times k}$ and $C \in \mathbb{R}^{k \times n}$ such that the Frobenius norm between $X$ and $BC$ is minimized. If no further restrictions are placed on the matrices $B$ and $C$, this problem can be solved optimally by computing the singular value decomposition. However, imposing restrictions on $B$ and $C$ can lead to factorizations which are more desirable for reasons such as interpretability and sparsity. One of the most common restrictions is non-negative matrix factorization (NMF), requiring $B$ and $C$ to consist only of non-negative entries (see [Berry et al., 2007] for a survey). Practically, NMF has seen widespread usage as it often produces nice factorizations that are frequently sparse. Typically NMF is accomplished by applying local search heuristics, and while NMF can be solved exactly in certain cases (see [Arora et al., 2016]), in general NMF is not only NP-hard [Vavasis, 2009] but also d-SUM-hard [Arora et al., 2016].

One drawback of factorizations such as SVD or NMF is that they can represent the data using a basis that may have no clear relation to the data. CU decompositions [Mahoney and Drineas, 2009] address this by requiring the basis to consist of input points. While it appears that the hardness of this problem has not been resolved, approximate solutions are known. Most notable is the additive approximation of Frieze et al. [2004], though more recently there have been advances on the multiplicative front [Drineas et al., 2008, Çivril and Magdon-Ismail, 2012, Guruswami and Sinop, 2012]. Similar restrictions have also been considered for NMF. Donoho and Stodden [2003] introduced a separability

assumption for NMF, and Arora et al. [2016] showed that a NMF can be computed in polynomial time under this assumption. Various other methods have since been proposed for NMF under the separability (or near separability) assumption [Recht et al., 2012, Kumar et al., 2013, Benson et al., 2014, Gillis and Vavasis, 2014, Zhou et al., 2014, Kumar and Sindhwani, 2015]. The separability assumption requires that there exists a subset $S$ of the columns of $X$ such that $X = X_S C$ for some nonnegative matrix $C$. This assumption can be restrictive in practice, e.g., when an exact subset does not exist but a close approximate subset does, i.e., $X \approx X_S C$. To our knowledge, no exact or approximate polynomial time algorithms have been proposed for the general problem of computing a NMF under only the restriction that the columns must be selected from those of $X$.

In this work, we fill this gap by arguing that a simple greedy algorithm can be used to provide a polynomial time $\varepsilon$-approximation algorithm for NMF under the column subset restriction. Note that the separability assumption is not required here: our theoretical analysis bounds the error of our selected columns versus the best possible columns that could have been chosen. The algorithm is based off of recent work on fast algorithms for approximately computing the convex hull of a set of points [Blum et al., 2016]. As in previous approaches [Donoho and Stodden, 2003, Kumar et al., 2013], we formulate restricted NMF geometrically as finding a subset, $S$, of the columns of the matrix $X$ whose conic hull, the set of all nonnegative combinations of columns of $S$, well-approximates the conic hull of $X$. Using gnomonic projection, we reduce the conic hull problem to a convex hull problem and then apply the greedy strategy of Blum et al. [2016] to compute the convex hull of the projected points. Given a set of points $P$ in $\mathbb{R}^m$, the convex hull of $S \subseteq P$, denoted Convex$(S)$, is said to $\varepsilon$-approximate Convex$(P)$ if the Hausdorff distance between Convex$(S)$ and Convex$(P)$ is at most $\varepsilon \cdot diameter(P)$. For a fixed $\varepsilon > 0$, suppose the minimum sized subset of $P$ whose convex hull $\varepsilon$-approximates the convex hull of $P$ has size $k$, then Blum et al. [2016] show that a simple greedy algorithm gives an $\varepsilon' = O(\varepsilon^{1/3})$ approximation using at most $k' = O(k/\varepsilon^{2/3})$ points of $P$, with an efficient $O(nc(m + c/\varepsilon^2 + c^2))$ running time, where $c = O(k_{opt}/\varepsilon^{2/3})$. By careful analysis, we show that our reduction achieves the same guarantees for the conic problem. (Note Blum et al. [2016] present other trade-offs between $k'$ and $\varepsilon'$, which we argue carry to the conic case as well). Significantly, $k'$ and $\varepsilon'$ are independent of $n$ and $m$, making this algorithm desirable for large high dimensional point sets. Note that our bounds on the approximation quality and the number of points do not explicitly depend on the dimension as they are relative to the size of the optimal solution, which itself may or may not depend on dimension. Like the X-RAY algorithm [Kumar et al., 2013], our algorithm is easy to parallelize, allowing it to be applied to large-scale problems.

In addition to the above $\varepsilon$-approximation algorithm, we also present two additional theoretical results of independent interest. The first theoretical contribution provides justification for empirical observations about the sparsity of NMF [Lee and Seung, 1999, Ding et al., 2010]. Due to the high dimensional nature of many data sets, there is significant interest in sparse representations requiring far fewer points than the dimension. Our theoretical justification for sparsity is based on Carathéodory's theorem: any point $q$ in the convex hull of $P$ can be expressed as a convex combination of at most $m + 1$ points from $P$. This is tight in the worst case for exact representation, however the approximate Carathéodory theorem [Clarkson, 2010, Barman, 2015] states there is a point $q'$ which is a convex combination of $O(1/\varepsilon^2)$ points of $P$ (i.e., independent of $n$ and $m$) such that $||q - q'|| \leq \varepsilon \cdot diameter(P)$. This result has a long history with significant implications in machine learning, e.g., relating to the analysis of the perceptron algorithm [Novikoff, 1962], though the clean geometric statement of this theorem appears to be not well known outside the geometry community. Moreover, this approximation is easily computable with a greedy algorithm (e.g., [Blum et al., 2016]) similar to the Frank-Wolfe algorithm. The analogous statement for the linear case does not hold, so it is not immediately obvious whether such an approximate Carathéodory theorem should hold for the conic case, a question which we answer in the affirmative. As a second theoretical contribution, we address the question of whether or not the convex/conic hull problems are actually hard, i.e., whether approximations are actually necessary. We answer this question for both problems in the affirmative, resolving an open question of Blum et al. [2016], by showing both that the conic and convex problems are d-SUM-hard.

Finally, we evaluate the performance of the greedy algorithms for computing the convex and conic hulls on a variety of feature selection tasks against existing methods. We observe that, both the conic and convex algorithms perform well for a variety of feature selection tasks, though, somewhat surprisingly, the convex hull algorithm, for which previously no experimental results had been

produced, yields consistently superior results on text datasets. We use our theoretical results to provide intuition for these empirical observations.

## 2 Preliminaries

Let $P$ be a point set in $\mathbb{R}^m$. For any $p \in P$, we interchangeably use the terms vector and point, depending on whether or not we wish to emphasize the direction from the origin. Let $\mathsf{ray}(p)$ denote the unbounded ray passing through $p$, whose base lies at the origin. Let $\mathsf{unit}(p)$ denote the unit vector in the direction of $p$, or equivalently $\mathsf{unit}(p)$ is the intersection of $\mathsf{ray}(p)$ with the unit hypersphere $\mathbb{S}^{(m-1)}$. For any subset $X = \{x_1, \ldots, x_k\} \subseteq P$, $\mathsf{ray}(X) = \{\mathsf{ray}(x_1), \ldots, \mathsf{ray}(x_k)\}$ and $\mathsf{unit}(X) = \{\mathsf{unit}(x_1), \ldots, \mathsf{unit}(x_k)\}$.

Given points $p, q \in P$, let $\mathsf{d}(p, q) = ||p-q||$ denote their Euclidean distance, and let $\langle p, q \rangle$ denote their dot product. Let $\mathsf{angle}(\mathsf{ray}(p), \mathsf{ray}(q)) = \mathsf{angle}(p, q) = \cos^{-1}(\langle \mathsf{unit}(p), \mathsf{unit}(q) \rangle)$ denote the angle between the rays $\mathsf{ray}(p)$ and $\mathsf{ray}(q)$, or equivalently between vectors $p$ and $q$. For two sets, $P, Q \subseteq \mathbb{R}^m$, we write $\mathsf{d}(P, Q) = \min_{p \in P, q \in Q} \mathsf{d}(p, q)$ and for a single point $q$ we write $\mathsf{d}(q, P) = \mathsf{d}(\{q\}, P)$, and the same definitions apply to $\mathsf{angle}()$.

For any subset $X = \{x_1, \ldots, x_k\} \subseteq P$, let $\mathsf{Convex}(X) = \{\sum_i \alpha_i x_i \mid \alpha_i \geq 0, \sum_i \alpha_i = 1\}$ denote the convex hull of $X$. Similarly, let $\mathsf{Conic}(X) = \{\sum_i \alpha_i x_i \mid \alpha_i \geq 0\}$ denote the conic hull of $X$ and $\mathsf{DualCone}(X) = \{z \in X \mid \langle x, z \rangle \geq 0 \; \forall x \in X\}$ the dual cone. For any point $q \in \mathbb{R}^m$, the *projection* of $q$ onto $\mathsf{Convex}(X)$ is the closest point to $q$ in $\mathsf{Convex}(X)$, $\mathsf{proj}(q) = \mathsf{proj}(q, \mathsf{Convex}(X)) = \arg\min_{x \in \mathsf{Convex}(X)} \mathsf{d}(q, x)$. Similarly the *angular projection* of $q$ onto $\mathsf{Conic}(X)$ is the angularly closest point to $q$ in $\mathsf{Conic}(X)$, $\mathsf{aproj}(q) = \mathsf{aproj}(q, \mathsf{Conic}(X)) = \arg\min_{x \in \mathsf{Conic}(X)} \mathsf{angle}(q, x)$. Note that angular projection defines an entire ray of $\mathsf{Conic}(X)$, rather than a single point, which without loss of generality we choose the point on the ray minimizing the Euclidean distance to $q$. In fact, abusing notation, we sometimes equivalently view $\mathsf{Conic}(X)$ as a set of rays rather than points, in which case $\mathsf{aproj}(\mathsf{ray}(q)) = \mathsf{aproj}(q)$ is the entire ray.

For $X \subset \mathbb{R}^m$, let $\Delta = \Delta_X = \max_{p,q \in X} \mathsf{d}(p, q)$ denote the *diameter* of $X$. The *angular diameter* of $X$ is $\phi = \phi_X = \max_{p,q \in X} \mathsf{angle}(p, q)$. Similarly $\phi_X(q) = \max_{p \in X} \mathsf{angle}(p, q)$ denotes the angular radius of the minimum radius cone centered around the ray through $q$ and containing all of $P$.

**Definition 2.1.** Consider a subset $X$ of a point set $P \subset \mathbb{R}^m$. $X$ is an $\varepsilon$-***approximation*** to $\mathsf{Convex}(P)$ if $d_{convex}(X, P) = \max_{p \in \mathsf{Convex}(P)} \mathsf{d}(p, \mathsf{Convex}(X)) \leq \varepsilon\Delta$. Note $d_{convex}(X, P)$ is the Hausdorff distance between $\mathsf{Convex}(X)$ and $\mathsf{Convex}(P)$. Similarly $X$ is an $\varepsilon$-*approximation* to $\mathsf{Conic}(P)$ if $d_{conic}(X, P) = \max_{p \in \mathsf{Conic}(P)} \mathsf{angle}(p, \mathsf{Conic}(X)) \leq \varepsilon\phi_P$.

Note that the definition of $\varepsilon$-*approximation* for $\mathsf{Conic}(P)$ uses angular rather than Euclidean distance in order to be defined for rays, i.e., scaling a point outside the conic hull changes its Euclidean distance but its angular distance is unchanged since its ray stays the same. Thus we find considering angles better captures what it means to approximate the conic hull than the distance based Frobenius norm which is often used to evaluate the quality of approximation for NMF.

As we are concerned only with angles, without loss of generality we often will assume that all points in the input set $P$ have been scaled to have unit length, i.e., $P = \mathsf{unit}(P)$. In our theoretical results, we will always assume that $\phi_P < \pi/2$. Note that if $P$ lies in the non-negative orthant, then for any strictly positive $q$, $\phi_P(q) < \pi/2$. In the case that the $P$ is not strictly inside the positive orthant, the points can be uniformly translated a small amount to ensure that $\phi_P < \pi/2$.

## 3 A Simple Greedy Algorithm

Let $P$ be a finite point set in $\mathbb{R}^m$ (with unit lengths). Call a point $p \in P$ *extreme* if it lies on the boundary of the conic hull (resp. convex hull). Observe that for any $X \subseteq P$, containing all the extreme points, it holds that $\mathsf{Conic}(X) = \mathsf{Conic}(P)$ (resp. $\mathsf{Convex}(X) = \mathsf{Convex}(P)$). Consider the simple greedy algorithm which builds a subset of points $S$, by iteratively adding to $S$ the point angularly furthest from the conic hull of the current point set $S$ (for the convex hull take the furthest point in distance). One can argue in each round this algorithm selects an extreme point, and thus can be used to find a subset of points whose hull captures that of $P$. Note if the hull is not degenerate, i.e.,

no point on the boundary is expressible as a combination of other points on the boundary, then this produces the minimum sized subset capturing $P$. Otherwise, one can solve a recursive subproblem as discussed by Kumar et al. [2013] to exactly recover $S$.

Here instead we consider finding a small subset of points (potentially much smaller than the number of extreme points) to approximate the hull. The question is then whether this greedy approach still yields a reasonable solution, which is not clear as there are simple examples showing the best approximate subset includes non-extreme points. Moreover, arguing about the conic approximation directly is challenging as it involves angles and hence spherical (rather than planar) geometry. For the convex case, Blum et al. [2016] argued that this greedy strategy does yield a good approximation. Thus we seek a way to reduce our conic problem to an instance of the convex problem, without introducing too much error in the process, which brings us to the *gnomonic projection*. Let $\mathsf{hplane}(q)$ be the hyperplane defined by the equation $\langle (q - x), q \rangle = 0$ where $q \in \mathbb{R}^m$ is a unit length normal vector. The *gnomonic projection* of $P$ onto $\mathsf{hplane}(q)$, is defined as $\mathsf{gp}^q(P) = \{\mathsf{ray}(P) \cap \mathsf{hplane}(q)\}$ (see Figure 3.1). Note that $\mathsf{gp}^q(q) = q$. For any point $x$ in $\mathsf{hplane}(q)$, the inverse gnomonic projection is $\mathsf{pg}^q(x) = \mathsf{ray}(x) \cap \mathbb{S}^{(m-1)}$. Similar to other work [Kumar et al., 2013], we allow projections onto any hyperplane tangent to the unit hypersphere with normal $q$ in the strictly positive orthant.

A key property of the gnomonic projection, is that the problem of finding the extreme points of the convex hull of the projected points is equivalent to finding the extreme points of the conic hull of $P$. (Additional properties of the gnomonic projection are discussed in the full version.) Thus the strategy to approximate the conic hull should now be clear. Let $P' = \mathsf{gp}^q(P)$. We apply the greedy strategy of Blum et al. [2016] to $P'$ to build a set of extreme points $S$, by iteratively adding to $S$ the point furthest from the convex hull of the current point set $S$. This procedure is shown in Algorithm 1.

We show that Algorithm 1 can be used to produce an $\varepsilon$-approximation to the restricted NMF problem. Formally, for $\varepsilon > 0$, let $opt(P, \varepsilon)$ denote any minimum cardinality subset $X \subseteq P$ which $\varepsilon$-approximates $\mathsf{Conic}(P)$, and let $k_{opt} = |opt(P, \varepsilon)|$. We consider the following problem.

**Problem 3.1.** *Given a set $P$ of $n$ points in $\mathbb{R}^m$ such that $\phi_P \leq \pi/2 - \gamma$, for a constant $\gamma > 0$, and a value $\varepsilon > 0$, compute $opt(P, \varepsilon)$.*

Alternatively one can fix $k$ rather than $\varepsilon$, defining $opt(P, k) = \arg\min_{X \subseteq P, |X| = k} d_{conic}(X, P)$ and $\varepsilon_{opt} = d_{conic}(opt(P, k), P)$. Our approach works for either variant, though here we focus on the version in Problem 3.1. Note the bounded angle assumption applies to any collection of points in the strictly positive orthant (a small translation can be used to ensure this for any nonnegative data set).

In this section we argue Algorithm 1 produces an $(\alpha, \beta)$-approximation to an instance $(P, \varepsilon)$ of Problem 3.1, that is a subset $X \subseteq P$ such that $d_{conic}(X, P) \leq \alpha$ and $|X| \leq \beta \cdot k_{opt} = \beta \cdot |opt(P, \varepsilon)|$.

For $\varepsilon > 0$, similarly define $opt^{convex}(P, \varepsilon)$ to be any minimum cardinality subset $X \subseteq P$ which $\varepsilon$-approximates $\mathsf{Convex}(P)$. Blum et al. [2016] gave $(\alpha, \beta)$-approximation for the following.

**Problem 3.2.** *Given a set $P$ of $n$ points in $\mathbb{R}^m$, and a value $\varepsilon > 0$, compute $opt^{convex}(P, \varepsilon)$.*

Note the proofs of correctness and approximation quality from Blum et al. [2016] for Problem 3.2 do not immediately imply the same results for using Algorithm 1 for Problem 3.1. To see this, consider any points $u, v$ on $\mathbb{S}^{(m-1)}$. Note the angle between $u$ and $v$ is the same as their geodesic distance on $\mathbb{S}^{(m-1)}$. Intuitively, we want to claim the geodesic distance between $u$ and $v$ is roughly the same as the Euclidean distance between $\mathsf{gp}^q(u)$ and $\mathsf{gp}^q(v)$. While this is true for points near $q$, as we move away from $q$ the correspondence breaks down (and is unbounded as you approach $\pi/2$). This non-uniform distortion requires care, and thus the proofs had to be moved to the full version.

Finally, observe that Algorithm 1, requires being able to compute the point furthest from the convex hull. To do so we use the (convex) approximate Carathéodory, which is both theoretically and practically very efficient, and produces provably sparse solutions. As a stand alone result, we first prove the conic analog of the approximate Carathéodory theorem. This result is of independent interest since it can be used to sparsify the returned solution from Algorithm 1, or any other algorithm.

### 3.1 Sparsity and the Approximate Conic Carathéodory Theorem

Our first result is a conic approximate Carathéodory theorem. That is, given a point set $P \subseteq \mathbb{R}^m$ and a query point $q$, then the angularly closest point to $q$ in $\mathsf{Conic}(P)$ can be approximately expressed as

---

**Algorithm 1:** Greedy Conic Hull

> **Data:** A set of $n$ points, $P$, in $\mathbb{R}^m$ such that $\phi_P < \pi/2$, a positive integer $k$, and a normal vector $q$ in $\mathsf{DualCone}(P)$.
>
> **Result:** $S \subseteq P$ such that $|S| = k$
>
> $Y \leftarrow \mathsf{gp}^q(P)$;
>
> Select an arbitrary starting point $p^0 \in Y$;
>
> $S \leftarrow \{p^0\}$;
>
> **for** $i$ = 2 to $k$ **do**
>
> > Select
> > $p^* \in \arg\max_{p \in Y} d_{convex}(p, S)$;
> > $S \leftarrow S \cup \{p^*\}$;

---

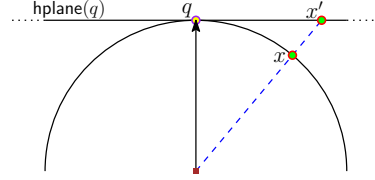

Figure 3.1: Side view of gnomonic projection.

a sparse combination of point from $P$. More precisely, one can compute a point $t$ which is a conic combination of $O(1/\varepsilon^2)$ points from $P$ such that $\mathsf{angle}(q, t) \le \mathsf{angle}(q, \mathsf{Conic}(P)) + \varepsilon\phi_P$.

The significance of this result is as follows. Recall that we seek a factorization $X \approx BC$, where the $k$ columns of $B$ are a subset of those from $X$ and the entries of $C$ are non-negative. Ideally each point in $X$ is expressed as a sparse combination from the basis $B$, that is each column of $C$ has very few non-zero entries. So suppose we are given any factorization $BC$, but $C$ is dense. Then no problem, just throw out $C$, and use our Carathéodory theorem to compute a new matrix $C'$ with sparse columns. Namely treat each column of $X$ as the query $q$ and run the theorem for the point set $P = B$, and then the non-zero entries of corresponding column of $C'$ are just the selected combination from $B$. Not only does this mean we can sparsify any solution to our NMF problem (including those obtained by other methods), but it also means conceptually that rather than finding a good pair $BC$, one only needs to focus on finding the subset $B$, as is done in Algorithm 1. Note that Algorithm 1 allows non-negative inputs in $P$ because $\phi_P < \pi/2$ ensures $P$ can be rotated into the positive orthant.

While it appears the conic approximate Carathéodory theorem had not previously been stated, the convex version has a long history (e.g., implied by [Novikoff, 1962]). The algorithm to compute this sparse convex approximation is again a simple and fast greedy algorithm, which roughly speaking is a simplification of the Frank-Wolfe algorithm for this particular problem. Specifically, to find the projection of $q$ onto $\mathsf{Convex}(P)$, start with any point $t_0 \in \mathsf{Convex}(P)$. In the $i$th round, find the point $p_i \in P$ most extreme in the direction of $q$ from $t_{i-1}$ (i.e., maximizing $\langle q - t_{i-1}, p_i \rangle$) and set $t_i$ to be the closest point to $q$ on the segment $t_{i-1}p_i$ (thus simplifying Frank Wolfe, as we ignore step size issues). The standard analysis of this algorithm (e.g., [Blum et al., 2016]) gives the following.

**Theorem 3.3 (Convex Carathéodory).** *For a point set $P \subseteq \mathbb{R}^m$, $\varepsilon > 0$, and $q \in \mathbb{R}^m$, one can compute, in $O\big(|P|\,m/\varepsilon^2\big)$ time, a point $t \in \mathsf{Convex}(P)$, such that $\mathsf{d}(q, t) \le \mathsf{d}(q, \mathsf{Convex}(P)) + \varepsilon\Delta$, where $\Delta = \Delta_P$. Furthermore, $t$ is a convex combination of $O(1/\varepsilon^2)$ points of $P$.*

Again by exploiting properties of the gnomonic projection we are able to prove a conic analog of the above theorem. Note for $P \subset \mathbb{R}^m$, $P$ is contained in the linear span of at most $m$ points from $P$, and similarly the exact Carathéodory theorem states any point $q \in \mathsf{Convex}(P)$ is expressible as a convex combination of at most $m + 1$ points from $P$. As the conic hull lies between the linear case (with all combinations) and the convex case (with non-negative combinations summing to one), it is not surprising an exact conic Carathéodory theorem holds. However, the linear analog of the approximate convex Caratheodory theorem does not hold, and so the following conic result is not a priori obvious.

**Theorem 3.4.** *Let $P \subset \mathbb{R}^m$ be a point set, let $q$ be such that $\phi_P(q) < \pi/2 - \gamma$ for some constant $\gamma > 0$, and let $\varepsilon > 0$ be a parameter. Then one can find, in $O(|P|m/\varepsilon^2)$ time, a point $t \in \mathsf{Conic}(P)$ such that $\mathsf{angle}(q, t) \le \mathsf{angle}(q, \mathsf{Conic}(P)) + \varepsilon\phi_P(q)$. Moreover, $t$ is a conic combination of $O(1/\varepsilon^2)$ points from $P$.*

Due to space constraints, the detailed proof of Theorem 3.4 appears in the full version. In the proof, the dependence on $\gamma$ is made clear but we make a remark about it here. If $\varepsilon$ is kept fixed, $\gamma$ shows up

in the running time roughly by a factor of $\tan^2(\pi/2 - \gamma)$. Alternatively, if the running time is fixed, the approximation error will roughly depend on the factor $1/\tan(\pi/2 - \gamma)$.

We now give a simple example of a high dimensional point set which shows our bounded angle assumption is required for the conic Carathéodory theorem to hold. Let $P$ consist of the standard basis vectors in $\mathbb{R}^m$, let $q$ be the all ones vector, and let $\varepsilon$ be a parameter. Let $X$ be a subset of $P$ of size $k$, and consider $\mathsf{aproj}(q) = \mathsf{aproj}(q, X)$. As $P$ consists of basis vectors, each of which have all but one entry set to zero, $\mathsf{aproj}(q)$ will have at most $k$ non-zero entries. By the symmetry of $q$ it is also clear that all non-zero entries in $\mathsf{aproj}(q)$ should have the same value. Without loss of generality assume that this value is 1, and hence the magnitude of $\mathsf{aproj}(q)$ is $\sqrt{k}$. Thus for $\mathsf{aproj}(q)$ to be an $\varepsilon$-approximation to $q$, $\mathsf{angle}(\mathsf{aproj}(q), q) = \cos^{-1}(\frac{k}{\sqrt{k}\sqrt{m}}) = \cos^{-1}(\sqrt{k/m}) < \varepsilon$. Hence for a fixed $\varepsilon$, the number of points required to $\varepsilon$-approximate $q$ depends on $m$, while the conic Carathéodory theorem should be independent of $m$.

## 3.2 Approximating the Conic Hull

We now prove that Algorithm 1 yields an approximation to the conic hull of a given point set and hence an approximation to the nonnegative matrix factorization problem. As discussed above, previously Blum et al. [2016] provided the following $(\alpha, \beta)$-approximation for Problem 3.2.

**Theorem 3.5 ([Blum et al., 2016]).** *For a set $P$ of $n$ points in $\mathbb{R}^m$, and $\varepsilon > 0$, the greedy strategy, which iteratively adds the point furthest from the current convex hull, gives a $((8\varepsilon^{1/3} + \varepsilon)\Delta, O(1/\varepsilon^{2/3}))$-approximation to Problem 3.2, and has running time $O(nc(m + c/\varepsilon^2 + c^2))$ time, where $c = O(k_{opt}/\varepsilon^{2/3})$.*

Our second result, is a conic analog of the above theorem.

**Theorem 3.6.** *Given a set $P$ of $n$ points in $\mathbb{R}^m$ such that $\phi_P \le \frac{\pi}{2} - \gamma$ for a constant $\gamma > 0$, and a value $\varepsilon > 0$, Algorithm 1 gives an $((8\varepsilon^{1/3} + \varepsilon)\phi_P, O(1/\varepsilon^{2/3}))$-approximation to Problem 3.1, and has running time $O(nc(m + c/\varepsilon^2 + c^2))$, where $c = O(k_{opt}/\varepsilon^{2/3})$.*

Bounding the approximation error requires carefully handling the distortion due to the gnomonic project, and the details are presented in the full version. Additionally, Blum et al. [2016] provide other $(\alpha, \beta)$-approximations, for different values of $\alpha$ and $\beta$, and in the full version these other results are also shown to hold for the conic case.

# 4 Hardness of the Convex and Conic Problems

This section gives a reduction from d-SUM to the convex approximation of Problem 3.2, implying it is d-SUM-hard. In the full version a similar setup is used to argue the conic approximation of Problem 3.1 is d-SUM-hard. Actually if Problem 3.1 allowed instances where $\phi_P = \pi/2$ the reduction would be virtually the same. However, arguing that the problem remains hard under our requirement that $\phi_P \le \pi/2 - \gamma$, is non-trivial and some of the calculations become challenging and lengthy. The reductions to both problems are partly inspired by Arora et al. [2016]. However, here, we use the somewhat non-standard version of d-SUM where repetitions are allowed as described below.

**Problem 4.1 (d-SUM).** *In the d-SUM problem we are given a set $S = \{s_1, s_2, \cdots, s_N\}$ of $N$ values, each in the interval $[0, 1]$, and the goal is to determine if there is a set of $d$ numbers (not necessarily distinct) whose sum is exactly $d/2$.*

It was shown by Patrascu and Williams [2010] that if d-SUM can be solved in $N^{o(d)}$ time then 3-SAT has a sub-exponential time algorithm, i.e., that the Exponential Time Hypothesis is false.

**Theorem 4.2 (d-SUM-hard).** *Let $d < N^{0.99}$, $\delta < 1$. If d-SUM on $N$ numbers of $O(d \log(N))$ bits can be solved in $O(N^{\delta d})$ time, then 3-SAT on $n$ variables can be solved in $2^{o(n)}$ time.*

We will prove the following decision version of Problem 3.2 is d-SUM-hard. Note in this section the dimension will be denoted by $d$ rather than $m$, as this is standard for d-SUM reductions.

**Problem 4.3.** *Given a set $P$ of $n$ points in $\mathbb{R}^d$, a value $\varepsilon > 0$, and an integer $k$, is there a subset $X \subseteq P$ of $k$ points such that $d_{convex}(X, P) \leq \varepsilon \Delta$, where $\Delta$ is the diameter of $P$.*

Given an instance of d-SUM with $N$ values $S = \{s_1, s_2, \cdots, s_N\}$ we construct an instance of Problem 4.3 where $P \subset \mathbb{R}^{d+2}$, $k = d$, and $\varepsilon = 1/3$ (or any sufficiently small value). The idea is to create $d$ clusters each containing $N$ points corresponding to a choice of one of the $s_i$ values. The clusters are positioned such that exactly one point from each cluster must be chosen. The $d + 2$ coordinates are labeled $a_i$ for $i \in [d]$, $w$, and $v$. Together, $a_1, \cdots, a_d$ determine the cluster. The $w$ dimension is used to compute the sum of the chosen $s_i$ values. The $v$ dimension is used as a threshold to determine whether d-SUM is a yes or no instance to Problem 4.3. Let $w(p_j)$ denote the $w$ value of an arbitrary point $p_j$.

We assume $d \geq 2$ as d-SUM is trivial for $d = 1$. Let $e_1, e_2, \cdots, e_d \in \mathbb{R}^d$ be the standard basis in $\mathbb{R}^d$, $e_1 = (1, \cdots, 0)$, $e_2 = (0, 1, \cdots, 0), \ldots$, and $e_d = (0, \cdots, 1)$. Together they form the unit $d$-simplex, and they define the $d$ clusters in the construction. Finally, let $\Delta^* = \sqrt{2 + (\varepsilon s_{max} - \varepsilon s_{min})^2}$ be a constant where $s_{max}$ and $s_{min}$ are, respectively, the maximum and minimum values in $S$.

**Definition 4.4.** The set of points $P \subset \mathbb{R}^{d+2}$ are the following

> $p_j^i$ points: For each $i \in [d]$, $j \in [N]$, set $(a_1, \cdots, a_d) = e_i$, $w = \varepsilon s_j$ and $v = 0$
> $q$ point: For each $i \in [d]$, $a_i = 1/d$, $w = \varepsilon/2$, $v = 0$
> $q'$ point: For each $i \in [d]$, $a_i = 1/d$ and $w = \varepsilon/2$, $v = \varepsilon \Delta^*$

**Lemma 4.5 (Proof in full version).** *The diameter of $P$, $\Delta_P$, is equal to $\Delta^*$.*

We prove completeness and soundness of the reduction. Below $P^i = \cup_j p_j^i$ denotes the $i$th cluster.

**Observation 4.6.** *If $\max_{p \in P} d(p, \mathsf{Convex}(X)) \leq \varepsilon \Delta$, then $d_{convex}(X, P) \leq \varepsilon \Delta$:* For point sets $A$ and $B = \{b_1, \ldots, b_m\}$, if we fix $a \in \mathsf{Convex}(A)$, then for any $b \in \mathsf{Convex}(B)$ we have $||a - b|| = ||a - \sum_i \alpha_i b_i|| = ||\sum_i \alpha_i (a - b_i)|| \leq \sum_i \alpha_i ||a - b_i|| \leq \max_i ||a - b_i||$.

**Lemma 4.7 (Completeness).** *If there is a subset $\{s_{k_1}, s_{k_2}, \cdots, s_{k_d}\}$ of $d$ values (not necessarily distinct) such that $\sum_{i \in [d]} s_{k_i} = d/2$, then the above described instance of Problem 4.3 is a true instance, i.e. there is a $d$ sized subset $X \subseteq P$ with $d_{convex}(X, P) \leq \varepsilon \Delta$.*

*Proof:* For each value $s_{k_i}$ consider the point $x_i = (e_i, \varepsilon \cdot s_{k_i}, 0)$, which by Definition 4.4 is a point in $P$. Let $X = \{x_1, \ldots, x_d\}$. We now prove $\max_{p \in P} d(p, \mathsf{Convex}(X)) \leq \varepsilon \Delta$, which by Observation 4.6 implies that $d_{convex}(X, P) \leq \varepsilon \Delta$.

First observe that for any $p_j^i$ in $P$, $d(p_j^i, x_i) = \sqrt{(w(p_j^i) - w(x_i))^2} \leq |\varepsilon s_j - \varepsilon s_{k_i}| \leq \varepsilon \Delta$. The only other points in $P$ are $q$ and $q'$. Note that $d(q, q') = \varepsilon \Delta^* = \varepsilon \Delta$ from Lemma 4.5. Thus if we can prove that $q \in \mathsf{Convex}(X)$ then we will have shown $\max_{p \in P} d(p, \mathsf{Convex}(X)) \leq \varepsilon \Delta$. Specifically, we prove that the convex combination $x = \frac{1}{d} \sum_i^d x_i$ is the point $q$. As $X$ contains exactly one point from each set $P^i$, and in each such set all points have $a_i = 1$ and all other $a_j = 0$, it holds that $x$ has $1/d$ for all the $a$ coordinates. All points in $X$ have $v = 0$ and so this holds for $x$ as well. Thus we only need to verify that $w(x) = w(q) = \varepsilon/2$, for which we have $w(x) = \frac{1}{d} \sum_i w(x_i) = \frac{1}{d} \sum_i \varepsilon s_{k_i} = \frac{1}{d}(\varepsilon d/2) = \varepsilon/2$. ∎

Proving soundness requires some helper lemmas. Note that in the above proof we constructed a solution to Problem 4.3 that selected exactly one point from each cluster $P^i$. We now prove that this is a required property.

**Lemma 4.8 (Proof in full version).** *Let $P \subset \mathbb{R}^{d+2}$ be as defined above, and let $X \subseteq P$ be a subset of size $d$. If $d_{convex}(X, P) \leq \varepsilon \Delta$, then for all $i$, $X$ contains exactly one point from $P^i$.*

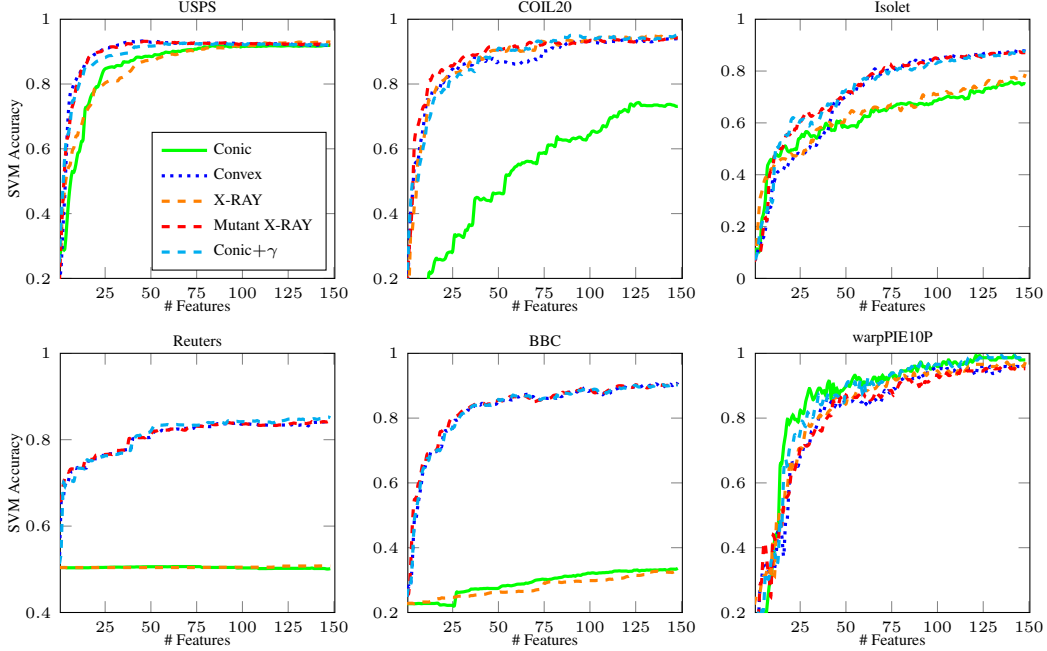

Figure 4.1: Experimental results for feature selection on six different data sets. Best viewed in color.

**Lemma 4.9 (Proof in full version).** *If $d_{convex}(X, P) \leq \varepsilon\Delta$, then $q \in \mathsf{Convex}(X)$ and moreover $q = \frac{1}{d}\sum_{x_i \in X} x_i$.*

**Lemma 4.10 (Soundness).** *Let $P$ be an instance of Problem 4.3 generated from a **d-SUM** instance $S$, as described in Definition 4.4. If there is a subset $X \subseteq P$ of size $d$ such that $d_{convex}(X, P) \leq \varepsilon\Delta$, then there is a choice of $d$ values from $S$ that sum to exactly $d/2$.*

*Proof:* From Lemma 4.8 we know that $X$ consist of exactly one point from each cluster $P^i$. Thus for each $x_i \in X$, $w(x_i) = \varepsilon s_{k_i}$ for some $s_{k_i} \in S$. By Lemma 4.9, $q = \frac{1}{d}\sum_i x_i$, which implies $w(q) = \frac{1}{d}\sum_i w(x_i)$. By Definition 4.4 $w(q) = \varepsilon/2$, which implies $\varepsilon/2 = \frac{1}{d}\sum_i w(x_i) = \frac{1}{d}\sum_i \varepsilon s_{k_i}$. Thus we have a set $\{s_{k_1}, \ldots, s_{k_d}\}$ of $d$ values from $S$ such that $\sum_i s_{k_i} = d/2$. ∎

Lemma 4.7 and Lemma 4.10 immediately imply the following.

**Theorem 4.11.** *For point sets in $\mathbb{R}^{d+2}$, Problem 4.3 is **d-SUM**-hard.*

# 5    Experimental Results

We report an experimental comparison of the proposed greedy algorithm for conic hulls, the greedy algorithm for convex hulls (the conic hull algorithm without the projection step) [Blum et al., 2016], the X-RAY (max) algorithm [Kumar et al., 2013], a modified version of X-RAY, dubbed mutant X-RAY, which simply selects the point furthest away from the current cone (i.e., with the largest residual), and a $\gamma$-shifted version of the conic hull algorithm described below. Other methods such as Hottopixx [Recht et al., 2012, Gillis and Luce, 2014] and SPA [Gillis and Vavasis, 2014] were not included due to their similar performance to the above methods. For our experiments, we considered the performance of each of the methods when used to select features for a variety of SVM classification tasks on various image, text, and speech data sets including several from the Arizona State University feature selection repository [Li et al., 2016] as well as the UCI Reuters dataset and the BBC News dataset [Greene and Cunningham, 2006]. The Reuters and BBC text datasets are represented using the TF-IDF representation. For the Reuters dataset, only the ten most frequent

topics were used for classification. In all datasets, columns (corresponding to features) that were identically equal to zero were removed from the data matrix.

For each problem, the data is divided using a 30/70 train/test split, the features are selected by the indicated method, and then an SVM classifier is trained using only the selected features. For the conic and convex hull methods, $\epsilon$ is set to $0.1$. The accuracy (percent of correctly classified instances) is plotted versus the number of selected features for each method in Figure 4.1. Additional experimental results can be found in the full version. Generally speaking, the convex, mutant X-RAY, and shifted conic algorithms seem to consistently perform the best on the tasks. The difference in performance between convex and conic is most striking on the two text data sets Reuters and BBC. In the case of BBC and Reuters, this is likely due to the fact that many of the columns of the TF-IDF matrix are orthogonal. We note that the quality of both X-RAY and conic is improved if thresholding is used when constructing the feature matrix, but they still seem to under perform the convex method for text datasets.

The text datasets are also interesting as not only do they violate the explicit assumption in our theorems that the angular diameter of the conic hull be strictly less than $\pi/2$, but that there are many such mutually orthogonal columns of the document-feature matrix. This observation motivates the $\gamma$-shifted version of the conic hull algorithm that simply takes the input matrix $X$ and adds $\gamma$ to all of the entries (essentially translating the data along the all ones vector) and then applies the conic hull algorithm. Let $1^{a,b}$ denote the $a \times b$ matrix of ones. After a nonnegative shift, the angular assumption is satisfied, and the restricted NMF problem is that of approximating $(X + \gamma 1^{m,n})$ as $(B + \gamma 1^{m,k})C$, where the columns of $B$ are again chosen from those of $X$. Under the Frobenus norm $||(X + \gamma 1^{m,n}) - (B + \gamma 1^{m,k})C||_2^2 = \sum_{i,j}(X_{ij} - B_{i,:}C_{:,j} + \gamma(1 - ||C_{:,j}||_1))^2$. As $C$ must be a nonnegative matrix, the shifted conic case acts like the original conic case plus a penalty that encourages the columns of $C$ to sum to one (i.e., it is a hybrid between the conic case and the convex case). The plots illustrate the performance of the $\gamma$-shifted conic hull algorithm for $\gamma = 10$. After the shift, the performance more closely matches that of the convex and mutant X-RAY methods on TF-IDF features.

Given these experimental results and the simplicity of the proposed convex and conic methods, we suggest that both methods should be added to practitioners' toolboxes. In particular, the superior performance of the convex algorithm on text datasets, compared to X-RAY and the conic algorithm, seems to suggest that these types of "convex" factorizations may be more desirable for TF-IDF features.

### Acknowledgments

Greg Van Buskirk and Ben Raichel were partially supported by NSF CRII Award-1566137. Nicholas Ruozzi was partially supported by DARPA Explainable Artificial Intelligence Program under contract number N66001-17- 2-4032 and NSF grant III-1527312

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
