[Supplementary Material · full.pdf]

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

# A  Gnomonic Projection

Here we provide some basic facts about the gnomonic projection. The lemma below follows by elementary trigonometry (see Figure 3.1).

**Observation A.1.** *For any $q \in \mathbb{S}^{(m-1)}$, the function $\mathsf{gp}^q(\cdot)$ defines a one to one correspondence between $\mathsf{hplane}(q)$ and the part of the hypersphere at angle $< \pi/2$ from $q$, i.e., the gnomonic projection projects the hemisphere defined by positive dot product with $q$ onto $\mathsf{hplane}(q)$.*

**Lemma A.2.** *Let $q, x$ be any two points on $\mathbb{S}^{(m-1)}$ such that $\mathsf{angle}(q, x) < \pi/2$. By the cosine formula for the dot product, $\mathsf{gp}^q(x) = \frac{x}{\langle x, q \rangle}$. Moreover, for any point $x'$ in $\mathsf{hplane}(q)$, by considering the triangle defined by the origin, $q$, and $x'$, the definition of the tangent function gives $\tan(\mathsf{angle}(q, x')) = \mathsf{d}(q, x')$.*

*In particular, for any point set $P \subseteq \mathbb{S}^{(m-1)}$ such that $\phi_P(q) < \pi/2$, $diameter(\mathsf{gp}^q(P)) \leq 2\tan(\phi_P(q))$, and notably this holds for $q = p$ for any $p \in P$ when $\phi_P < \pi/2$.*

As $\tan(\theta)$ is monotonically increasing for $0 \leq \theta < \pi/2$, we have the following corollary.

**Corollary A.3.** *Let $q, x, y$ be points on $\mathbb{S}^{(m-1)}$ such that $\mathsf{angle}(q, x), \mathsf{angle}(q, y) < \pi/2$. Then $\mathsf{angle}(q, x) \leq \mathsf{angle}(q, y)$ if and only if $\mathsf{d}(q, \mathsf{gp}^q(x)) \leq \mathsf{d}(q, \mathsf{gp}^q(y))$. Conversely, for any two points $x', y' \in \mathsf{hplane}(q)$, $\mathsf{d}(q, x') \leq \mathsf{d}(q, y')$ if and only if $\mathsf{angle}(q, \mathsf{pg}^q(x')) \leq \mathsf{angle}(q, \mathsf{pg}^q(y'))$.*

The proofs of the following lemmas are straightforward and are provided for completeness.

**Lemma A.4.** *Given a point set $P \subset \mathbb{R}^m$, $\mathsf{Conic}(P) = \mathsf{ray}(\mathsf{Convex}(P))$.*

*Proof:* First note that any $x \in \mathsf{Conic}(P)$ is a scaling of some point $z \in \mathsf{Convex}(P)$, i.e. $x = \lambda z$ for some value $\lambda$. To see this, let $x = \sum \alpha_i p_i$ such that $\alpha_i \geq 0$. Now let $\lambda = \sum \alpha_i$ and $z = \frac{1}{\lambda} \sum \alpha_i p_i$. The other direction can be shown in a similar way. ∎

**Lemma A.5.** *Given a point set $P \subset \mathbb{R}^m$ and a point $q \in \mathbb{S}^{(m-1)}$ such that $\phi_P(q) < \pi/2$, it holds that $\mathsf{Conic}(P) \cap \mathsf{hplane}(q) = \mathsf{Convex}(\mathsf{gp}^q(P))$*

*Proof:* Let $P = \{p_1, \ldots, p_n\}$. As $\phi_P(q) < \pi/2$, by *Observation A.1*, each $p_i$ defines a unique $p_i' = \mathsf{gp}^q(p_i)$ in $\mathsf{hplane}(q)$, and by Lemma A.2, $p_i' = p_i / \langle p_i, q \rangle$. Let $\mathsf{gp}^q(P) = P' = \{p_1', \ldots, p_n'\}$.

First we show $\mathsf{Conic}(P) \cap \mathsf{hplane}(q) \supseteq \mathsf{Convex}(P')$. Consider any point in $x' \in \mathsf{Convex}(P')$. Note that $x' = \sum_i \alpha_i p_i' = \sum_i \frac{\alpha_i}{\langle p_i, q \rangle} p_i$, where $\sum_i \alpha_i = 1$ and $\alpha_i \geq 0$ for all $i$. Thus $x' \in \mathsf{Conic}(P)$ as the coefficients $\frac{\alpha_i}{\langle p_i, q \rangle}$ are non-negative. Moreover, $x' \in \mathsf{hplane}(q)$ as it is a convex combination of points in $\mathsf{hplane}(q)$.

Now we argue $\mathsf{Conic}(P) \cap \mathsf{hplane}(q) \subseteq \mathsf{Convex}(P')$. Consider any point $x$ in $\mathsf{Convex}(P)$, that is $x = \sum_i \alpha_i p_i$, where $\sum_i \alpha_i = 1$ and $\alpha_i \geq 0$ for all $i$. First observe since $x$ is a convex combination of the $p_i$, each of which has positive dot product with $q$, by linearity of the dot product, so does $x$, and hence $\mathsf{gp}^q(x)$ is a well defined point on $\mathsf{hplane}(q)$. We now argue $\mathsf{gp}^q(x) \in \mathsf{Convex}(P')$, which by Lemma A.4 implies the claim as $\mathsf{gp}^q(x)$ is on $\mathsf{ray}(x)$ for $x \in \mathsf{Convex}(P)$.

$$\mathsf{gp}^q(x) = \frac{x}{\langle x, q \rangle} = \frac{\sum_i \alpha_i p_i}{\left\langle (\sum_j \alpha_j p_j), q \right\rangle} = \frac{\sum_i \alpha_i p_i}{\sum_j \alpha_j \langle p_j, q \rangle} = \frac{\sum_i \alpha_i \langle p_i, q \rangle p_i'}{\sum_j \alpha_j \langle p_j, q \rangle} = \sum_i \frac{\alpha_i \langle p_i, q \rangle}{\sum_j \alpha_j \langle p_j, q \rangle} p_i'.$$

Thus $\mathsf{gp}^q(x) \in \mathsf{Convex}(P')$ as the coefficients are non-negative and $\sum_i \frac{\alpha_i \langle p_i, q \rangle}{\sum_j \alpha_j \langle p_j, q \rangle} = 1$. ∎

# B  Proof of Theorem 3.4

**Theorem B.1 (Restatement of Theorem 3.4).** *Let $P \subset \mathbb{R}^m$ be a point set, let $q \in \mathbb{S}^{(m-1)}$ be such that $\phi_P(q) < \pi/2 - \gamma$ for some constant $\gamma > 0$, and let $\varepsilon > 0$ be a parameter. Then one can find, in $O(|P|m/\varepsilon^2)$ time, a point $t \in \mathsf{Conic}(P)$ such that $\mathsf{angle}(q, t) \leq \mathsf{angle}(q, \mathsf{Conic}(P)) + \varepsilon \phi_P(q)$. Moreover, $t$ is a conic combination of $O(1/\varepsilon^2)$ points from $P$.*

*Proof:* First compute the set $\mathsf{gp}^q(P)$. (Note that since $\phi_P(q) < \pi/2$, by Observation A.1, every point $p \in P$ defines a unique point $\mathsf{gp}^q(p) \in \mathsf{hplane}(q)$.) Let $\varepsilon' = \varepsilon\phi_P(q)/(2\tan(\phi_P(q)))$. Note that $\phi_P(q) < \pi/2 - \gamma$ implies $c \le \phi_P(q)/(2\tan(\phi_P(q))) \le 1/2$, for some positive constant $c$ depending only on the constant $\gamma$, and thus $\varepsilon' = \Theta(\varepsilon)$.

Let $opt' \in \mathsf{Convex}(\mathsf{gp}^q(P))$ be a point in the convex hull of $\mathsf{gp}^q(P)$ that is closest to $q$. By Theorem 3.3, one can compute a point $t \in \mathsf{Convex}(\mathsf{gp}^q(P))$ such that $\mathsf{d}(q,t) \le \mathsf{d}(q,opt') + \varepsilon'\Delta$, where $\Delta = diameter(\mathsf{gp}^q(P))$, in $O(|P|m/\varepsilon'^2) = O(|P|m/\varepsilon^2)$ time, which is a convex combination of a subset $Z$ of $O(1/\varepsilon'^2) = O(1/\varepsilon^2)$ points of $\mathsf{gp}^q(P)$. We return $\mathsf{pg}^q(Z)$ as our solution. (Note $\mathsf{pg}^q(Z)$ is returned instead of $Z$, only because the theorem requires returning a subset of $P$.) Thus the running time and size of $\mathsf{pg}^q(Z)$ in the theorem statement hold, and what remains is proving $t$ yields the desired approximation.

By Lemma A.2, $\Delta \le 2\tan(\phi_P(q))$. By the same lemma, for any $x \in \mathsf{Convex}(\mathsf{gp}^q(P))$, $\mathsf{angle}(x,q) = \tan^{-1}(\mathsf{d}(x,q))$. Since $\tan^{-1}(\theta)$ is monotonically increasing for $0 \le \theta \le \pi/2$,

$$\mathsf{angle}(q,t) = \tan^{-1}(\mathsf{d}(q,t)) \le \tan^{-1}(\mathsf{d}(q,opt') + \varepsilon'\Delta)$$
$$\le \tan^{-1}(\mathsf{d}(q,opt') + 2\varepsilon' \cdot \tan(\phi_P(q))) = \tan^{-1}(\mathsf{d}(q,opt') + \varepsilon\phi_P(q)).$$

Furthermore, because $\tan^{-1}(\theta)$ is concave between $0 \le \theta \le \pi/2$ and $\frac{d}{d\theta}\tan^{-1}(\theta)|_0 = 1$,

$$\tan^{-1}(\mathsf{d}(q,opt') + \varepsilon\phi_P(q)) \le \tan^{-1}(\mathsf{d}(q,opt')) + \varepsilon\phi_P(q) = \mathsf{angle}(q,opt') + \varepsilon\phi_P(q)$$

Let $opt \in \mathsf{Conic}(P)$ be a point in the conic hull of $P$ that has smallest angle to $q$. To complete the proof, observe that Lemma A.5 implies both that $\mathsf{gp}^q(opt) \in \mathsf{Convex}(\mathsf{gp}^q(P))$ and $t \in \mathsf{Conic}(P)$. By definition $\mathsf{d}(opt',q) \le \mathsf{d}(\mathsf{gp}^q(opt),q)$, and thus by Corollary A.3, $\mathsf{angle}(opt',q) \le \mathsf{angle}(opt,q)$ (which actually implies $\mathsf{angle}(opt',q) = \mathsf{angle}(opt,q)$). Combining with the above inequality gives $\mathsf{angle}(q,t) \le \mathsf{angle}(q,opt') + \varepsilon\phi_P(q) \le \mathsf{angle}(q,opt) + \varepsilon\phi_P(q)$ ∎

## C  Proof of Theorem 3.6

The following helper lemma is crucial in bounding the approximation quality of our algorithm.

**Lemma C.1.** *Let $P \subset \mathbb{S}^{(m-1)}$ be a point set on the hypersphere such that $\phi_P \le \frac{\pi}{2} - \gamma$ for some constant $\gamma > 0$, and let $X \subseteq P$ be a non-empty subset. For any $q \in X$ and $p \in P$, let $y = \mathsf{pg}^q(\mathsf{proj}(\mathsf{gp}^q(p), \mathsf{Convex}(\mathsf{gp}^q(X))))$ be the projection of $\mathsf{gp}^q(p)$ onto $\mathsf{Convex}(\mathsf{gp}^q(X))$ pulled back to $\mathbb{S}^{(m-1)}$. Then it holds that $\mathsf{angle}(p,y) \le 3\ell$, where $\ell = d(\mathsf{gp}^q(p), \mathsf{gp}^q(y))$.*

*Proof:* First observe that if $y = q$, the angle between $p$ and $y$ is simply $\tan^{-1}(\ell) \le \ell < 3\ell$. Also, if $y = p$, i.e. $\mathsf{gp}^q(p)$ is in $\mathsf{Convex}(\mathsf{gp}^q(X))$, then $\mathsf{angle}(p,y) = 0$. So for the remainder of the proof we assume $y \neq q$ and $\mathsf{gp}^q(p) \notin \mathsf{Convex}(\mathsf{gp}^q(X))$.

We need to show some basic relations about these points and angles that are needed in the proof. Consider the spherical triangle defined by points $p, y, q$. Let $\beta = \mathsf{angle}(q,p)$, and let $\theta$ be the interior angle opposite the side $py$, see Figure C.1. First observe that $\ell \le \tan(\beta)$ as otherwise, $q$ would be closer to $\mathsf{gp}^q(p)$ than $\mathsf{gp}^q(y)$. Next, observe that $\mathsf{angle}(\mathsf{gp}^q(y),q) \le \mathsf{angle}(\mathsf{gp}^q(p),q)$. To see this, assume the contrary and note that we could then draw a perpendicular line from $\mathsf{gp}^q(p)$ to the line $q\,\mathsf{gp}^q(y)$. The point of intersection would then be closer to $\mathsf{gp}^q(p)$ than $\mathsf{gp}^q(y)$ which is a contradiction.

The last relation we need involves a new point, $z$, which we now define. Consider two circles on the sphere. The first is the great circle through $q$ and $y$. The second is the latitude circle at $p$ with respect to $q$ (i.e. $q$ acts as the north pole), which is the set of all points on the sphere with fixed angle $\beta = \mathsf{angle}(q,p)$ from $q$. Let $z$ be the intersection point of these two circles (technically there are two intersections, so take the one closer to $p$). See Figure C.1. Let $\theta_1 = \angle\mathsf{gp}^q(y)\mathsf{gp}^q(p)\mathsf{gp}^q(z)$ and $\theta_2 = \angle q\mathsf{gp}^q(z)\mathsf{gp}^q(p)$ (see Figure C.2) and note that $\theta_1 \le \theta_2$. Hence by the law of sines, we obtain $\mathsf{d}(\mathsf{gp}^q(y), \mathsf{gp}^q(z)) \le \ell$.

To prove the lemma, we invoke the spherical triangle inequality, that is $\mathsf{angle}(p,y) \le \mathsf{angle}(p,z) + \mathsf{angle}(z,y)$. We now upper bound $\mathsf{angle}(p,z)$ and $\mathsf{angle}(z,y)$ separately.

Figure C.1

Figure C.2

To bound $\mathsf{angle}(y, z)$, first observe that $q$, $y$, and $z$ are all on the same great circle, and so $\mathsf{angle}(q, z) = \mathsf{angle}(q, y) + \mathsf{angle}(y, z)$. By Lemma A.2,

$$\mathsf{angle}(y, z) = \mathsf{angle}(q, z) - \mathsf{angle}(q, y) = \tan^{-1}(d(\mathsf{gp}^q(z), q)) - \tan^{-1}(d(\mathsf{gp}^q(y), q))$$

$$= \tan^{-1}(d(\mathsf{gp}^q(y), q) + d(\mathsf{gp}^q(y), \mathsf{gp}^q(z))) - \tan^{-1}(d(\mathsf{gp}^q(y), q))$$

$$\leq \tan^{-1}(d(\mathsf{gp}^q(y), q)) + d(\mathsf{gp}^q(y), \mathsf{gp}^q(z)) - \tan^{-1}(d(\mathsf{gp}^q(y), q)) = d(\mathsf{gp}^q(y), \mathsf{gp}^q(z)) \leq \ell,$$

where the first inequality follows as $\tan^{-1}(\zeta)$ is concave for $0 \leq \zeta \leq \pi/2$ and $\frac{d}{d\zeta} \tan^{-1}(\zeta)|_0 = 1$ (i.e. the same argument from Theorem B.1).

To bound $\mathsf{angle}(p, z)$, first observe that $\beta = \mathsf{angle}(q, p) = \mathsf{angle}(q, z)$, that is we have an isosceles spherical triangle. Consider the corresponding isosceles triangle in $\mathsf{hplane}(q)$ defined by the points $q$, $\mathsf{gp}^q(p)$, and $\mathsf{gp}^q(z)$, see Figure C.2. Note that the interior spherical angle $\theta$ defined above is also the angle at $q$ in this planar triangle, that is $\theta = \angle \mathsf{gp}^q(p) \, q \, \mathsf{gp}^q(y)$. Also, $\theta < \pi/2$ as otherwise $\mathsf{d}(q, \mathsf{gp}^q(p)) \leq \mathsf{d}(\mathsf{gp}^q(p), \mathsf{gp}^q(y))$ contradicting the assumption $\mathsf{gp}^q(y) = \mathsf{proj}(p)$. By Lemma A.2, $d(\mathsf{gp}^q(p), q) = d(\mathsf{gp}^q(z), q) = \tan(\beta)$, so by the law cosines,

$$\cos(\theta) = \frac{\tan^2(\beta) + \tan^2(\beta) - d(\mathsf{gp}^q(p), \mathsf{gp}^q(z))^2}{2 \cdot \tan(\beta) \cdot \tan(\beta)} = 1 - \frac{d(\mathsf{gp}^q(p), \mathsf{gp}^q(z))^2}{2 \cdot \tan^2(\beta)}.$$

Going back to the sphere, by the spherical cosine law,

$$\cos(\mathsf{angle}(p, z)) = \cos(\beta)\cos(\beta) + \sin(\beta)\sin(\beta)\cos(\theta) =$$

$$\cos^2(\beta) + \sin^2(\beta) \cdot \left(1 - \frac{d(\mathsf{gp}^q(p), \mathsf{gp}^q(z))^2}{2 \cdot \tan^2(\beta)}\right) = 1 - \frac{\sin^2(\beta) \cdot d(\mathsf{gp}^q(p), \mathsf{gp}^q(z))^2}{2 \cdot \sin^2(\beta)/\cos^2 \beta}$$

$$= 1 - \cos^2(\beta) \cdot \frac{d(\mathsf{gp}^q(p), \mathsf{gp}^q(z))^2}{2} \geq 1 - 2\ell^2 \cdot \cos^2(\beta),$$

where the last inequality follows from $\mathsf{d}(\mathsf{gp}^q(p), \mathsf{gp}^q(z)) \leq \mathsf{d}(\mathsf{gp}^q(p), \mathsf{gp}^q(y)) + \mathsf{d}(\mathsf{gp}^q(y), \mathsf{gp}^q(z)) \leq 2\ell$, by the above relations. Remember that $\ell \leq \tan(\beta)$ and $\tan^{-1}(\cdot)$ is monotonically increasing. Therefore, as $\cos^{-1}(\cdot)$ is a monotonically decreasing function,

$$\mathsf{angle}(p, z) \leq \cos^{-1}(1 - 2\ell^2 \cdot \cos^2(\beta)) \leq \cos^{-1}(1 - 2\ell^2 \cdot \cos^2(\tan^{-1}(\ell))) = \cos^{-1}\left(1 - \frac{2\ell^2}{1 + \ell^2}\right).$$

We now show $\cos^{-1}(1 - 2\ell^2/(1 + \ell^2)) \leq 2\ell$. Consider the ratio $\frac{\cos^{-1}(1 - 2\ell^2/(1 + \ell^2))}{\ell}$ and observe that the derivative is negative for $\ell > 0$ ($\ell$ is always positive). Thus, $\frac{\cos^{-1}(1 - 2\ell^2/(1 + \ell^2))}{\ell}$ is increasing as $\ell$ approaches $0^+$. By L'Hopital's rule,

$$\lim_{\ell \to 0^+} \frac{\cos^{-1}(1 - 2\ell^2/(1 + \ell^2))}{\ell} = \lim_{\ell \to 0^+} \frac{2}{\ell + 1} = 2.$$

Therefore, the ratio is upper bounded by 2, and $\mathsf{angle}(p, z) \leq \cos^{-1}(1 - 2\ell^2/(1 + \ell^2)) \leq 2\ell$ holds. Note that because $2\ell^2/(1 + \ell^2) < 2$, the upper bound we obtained for $\mathsf{angle}(p, z)$, i.e. $\cos^{-1}(1 - 2\ell^2/(1 + \ell^2))$, is well defined for any value $\ell > 0$.

Finally, the spherical triangle inequality tell us our bounds $\mathsf{angle}(y, z) \leq \ell$ and $\mathsf{angle}(p, z) \leq 2\ell$, imply $\mathsf{angle}(p, y) \leq 3\ell$. ∎

We wish to prove the conic analog of the following theorem from [Blum et al., 2016], which is a more general form of Theorem 3.5 from Section 3.2.

**Theorem C.2 ([Blum et al., 2016]).** *Given a set $P$ of $n$ points in $\mathbb{R}^m$, and a value $\varepsilon > 0$, in polyno-mial one can compute:(i) an $(\varepsilon\Delta, O(m\log k_{opt}))$-approximation, (ii) a $((1+\delta)\varepsilon\Delta, O(\log(n)/(\varepsilon\delta)))$-approximation, and (iii) an $((8\varepsilon^{1/3}+\varepsilon)\Delta, O(1/\varepsilon^{2/3}))$-approximation to Problem 3.2. Moreover, for (iii), the run time is $O(nc(m+c/\varepsilon^2+c^2))$, where $c = O(k_{opt}/\varepsilon^{2/3})$.*

We are now ready to prove our main theorem, which is a more general form of Theorem 3.6. The proof will make use of the following simple observation about $d_{convex}(X, P)$ and $d_{conic}(X, P)$, which we formally state as it is also used in our d-SUM-harness proofs.

**Observation C.3.** *If $\max_{p\in P} \mathsf{d}(p, \mathsf{Convex}(X)) \leq \varepsilon\Delta$, then $d_{convex}(X, P) \leq \varepsilon\Delta$: For point sets $A$ and $B = \{b_1, \ldots, b_m\}$, if we fix $a \in \mathsf{Convex}(A)$, then for any $b \in \mathsf{Convex}(B)$ we have $||a-b|| = ||a - \sum_i \alpha_i b_i|| = ||\sum_i \alpha_i(a-b_i)|| \leq \sum_i \alpha_i ||a-b_i|| \leq \max_i ||a-b_i||$.*

*Similarly, if $\max_{p\in P} \mathsf{d}(p, \mathsf{Conic}(X)) \leq \varepsilon\phi_P$, then $d_{conic}(X, P) \leq \varepsilon\phi_P$: Again for sets $A$ and $B$ with $\phi_{A\cup B} < \pi/2$, fix $a \in \mathsf{Conic}(A)$ and $b \in \mathsf{Conic}(B)$. Lemma A.5 implies that $\mathsf{gp}^a(b)$ is a convex combination of $\mathsf{gp}^a(b_i)$, and thus the argument for the convex case implies $||a - \mathsf{gp}^a(b)|| \leq \max_i ||a - \mathsf{gp}^a(b_i)||$. Corollary A.3 then implies $\mathsf{angle}(a, b) \leq \max_i \mathsf{angle}(a, b_i)$.*

**Theorem C.4.** *Given a set $P$ of $n$ points in $\mathbb{R}^m$ such that $\phi_P \leq \frac{\pi}{2} - \gamma$ for a constant $\gamma > 0$, and a value $\varepsilon > 0$, in polynomial time one can compute: (i) an $(\varepsilon\phi_P, O(m\log k_{opt}))$-approximation, (ii) a $((1+\delta)\varepsilon\phi_P, O(\log(n)/(\varepsilon\delta)))$-approximation, and (iii) an $((8\varepsilon^{1/3}+\varepsilon)\phi_P, O(1/\varepsilon^{2/3}))$-approximation to Problem 3.1. Moreover, for case (iii), the run time is $O(nc(m+c/\varepsilon^2+c^2))$, where $c = O(k_{opt}/\varepsilon^{2/3})$.*

*Proof:* First note that as we are only concerned with angles, without loss of generality we assume $P \subset \mathbb{S}^{(m-1)}$. Set $\varepsilon' = c\cdot\varepsilon$ for a constant $c > 0$ to be determined shortly, and let $q = \mathsf{unit}(p)$ for an arbitrary point $p$ in $P$. Compute the set $P' = \mathsf{gp}^q(P)$, and consider the instance $(P', \varepsilon')$ of Problem 3.2. By Theorem C.2, in polynomial time we can compute a subset $X' \subseteq P'$ that is an $(\alpha, \beta)$-approximation to Problem 3.2 where $(\alpha, \beta)$ is either (i) $(\varepsilon'\Delta, O(m\log k_{opt}))$, (ii) $((1+\delta)\varepsilon'\Delta, O(\log(n)/(\varepsilon\delta)))$, or (iii) $((8(\varepsilon')^{1/3}+\varepsilon')\Delta, O(1/\varepsilon^{2/3}))$, where we used the fact that $\varepsilon' = \Theta(\varepsilon)$.

Note that $X = \mathsf{unit}(X')$ is a subset of $P$. To prove the theorem, we apply Lemma C.1 to the sets $X$ and $P$, which we now argue implies $X$ is a $(3\alpha, \beta)$-approximation to the instance $(P, \varepsilon)$ of Problem 3.1. First observe that in order to apply the lemma, $q$ must be in $X$, so if it is not already, simply add $q$ to $X$ as this does not asymptotically change $\beta$. By Definition 2.1, if $X'$ is an $(\alpha, \beta)$-approximation to an instance $(P', \varepsilon')$ of Problem 3.2, then for any point $\mathsf{gp}^q(p) \in P'$ the distance to its projection onto $\mathsf{Convex}(X')$, $y' = \mathsf{proj}(\mathsf{gp}^q(p), \mathsf{Convex}(X'))$, is at most $\alpha$. Thus setting $\ell = d(\mathsf{gp}^q(p), y')$, Lemma C.1 implies $\mathsf{angle}(p, \mathsf{pg}^q(y')) \leq 3\ell \leq 3\alpha$. Note by Lemma A.5, $\mathsf{pg}^q(y') \in \mathsf{Conic}(X)$, and so every point $p \in P$ has a point in $\mathsf{Conic}(X)$ at angle $\leq 3\alpha$, which by Observation C.3 implies $X$ is a $(3\alpha, \beta)$-approximation to Problem 3.1.

To complete the proof, we provide constants $c$ such that either (i) $3(c\varepsilon\Delta) \leq \varepsilon\phi_P$, or (ii) $3(1+\delta)c\varepsilon\Delta \leq (1+\delta)\varepsilon\phi_P$, or (iii) $3(8(c\varepsilon)^{1/3}+c\varepsilon)\Delta \leq (8\varepsilon^{1/3}+\varepsilon)\phi_P$. For case (i) and (ii), set $c = \frac{\phi_P}{6\tan(\phi_P)}$. Note that $c$ can indeed be treated as a positive constant because $\frac{\phi_P}{6\tan(\phi_P)}$ is always less than 1, and is bounded away from 0 by our assumption that $\phi_P \leq \pi/2 - \gamma$ for some constant $\gamma$ (and thus we still have $\varepsilon' = \Theta(\varepsilon)$). From Lemma A.2, $\Delta \leq 2\tan(\phi_P)$, and so $3(c\varepsilon\Delta) = \frac{3\varepsilon\Delta\phi_P}{6\tan(\phi_P)} \leq \frac{6\varepsilon\phi_P\tan(\phi_P)}{6\tan(\phi_P)} = \varepsilon\phi_P$, thus satisfying case (i). Similarly, $3(1+\delta)c\varepsilon\Delta \leq (1+\delta)\varepsilon\phi_P$, satisfying case (ii).

For case (iii) set $c = \left(\frac{\phi_P}{6\tan(\phi_P)}\right)^3$, which is still a constant. Since $\frac{\phi_P}{6\tan(\phi_P)} \leq 1$ we have $3(8(c\varepsilon)^{1/3}+c\varepsilon)\Delta \leq 3(8\left(\frac{\phi_P}{6\tan(\phi_P)}\right)\varepsilon^{1/3}+\left(\frac{\phi_P}{6\tan(\phi_P)}\right)^3\varepsilon)2\tan(\phi_P) \leq (8\varepsilon^{1/3}+\left(\frac{\phi_P}{6\tan(\phi_P)}\right)^2\varepsilon)\phi_P \leq (8\varepsilon^{1/3}+\varepsilon)\phi_P$. ∎

## D   Additional Hardness Results

### D.1   Problem 3.1 is d-SUM-hard

This section proves the following decision version of Problem 3.1 is d-SUM-hard. The approach is the same as in the convex reduction where we construct a point set $P$ with $d$ clusters each having $N$

points. However, we now require that $\phi_P \leq \pi/2 - \gamma$ to match the problem statement in Problem 3.1. Without this requirement, the reduction would be virtually the same as before, except with angles not distances. As a consequence of the bounded angle assumption, some of the calculations become very technically challenging.

**Problem D.1.** *Given a set $P$ of $n$ points in $\mathbb{R}^d$ such that $\phi_P \leq \pi/2 - \gamma$, a value $\varepsilon > 0$, and an integer $k$, is there a subset $X \subseteq P$ of $k$ points such that $d_{conic}(X, P) \leq \varepsilon \phi_P$, where $\phi_P$ is the angular diameter of $P$.*

Given an instance of d-SUM with $N$ values $S = \{s_1, s_2, \cdots, s_N\}$ we construct an instance of Problem D.1 where $P \subset \mathbb{R}^{d+2}$, $k = d$, and $\varepsilon = 1/7$. Let $u = (1, 1 \cdots, 1)$ be the all ones vector in $\mathbb{R}^d$, $e_i$ be the $i$th standard basis vector in $\mathbb{R}^d$, and $w^* = \tan(\varepsilon \cos^{-1}\left(\frac{d+2+1/5}{d+3}\right))$ be a constant. Note that setting $\varepsilon = 1/7$ still implies the general $\varepsilon$ case is hard, and moreover the problem remains hard even if we restrict to cases with $\varepsilon \geq 1/7$. In this section we assume $d \geq 4$, which is not necessary but it simplifies some of our later calculations.

**Definition D.2.** The set of points $P \subset \mathbb{R}^{d+2}$ are the following

$p_j^i$ points: For each $i \in [d]$, $j \in [N]$, set $(a_1, \cdots, a_d) = u + e_i$, $w = s_j \cdot w^*$ and $v = 0$
Let $R = \cup_{i,j} p_j^i$ be the set of all $p_j^i$ points

$q$ point: For each $i \in [d]$, $a_i = \frac{1+d}{d}$, $w = w^*/2$, $v = 0$

$q'$ point For each $i \in [d]$, $a_i = \frac{1+d}{d}$, $w = w^*/2$, $v = ||q|| \cdot \tan(\varepsilon \phi_R)$

Again, we let $P^i = \cup_j p_j^i$ denote the $i$th cluster. Observe that $w^*$ is the largest $w$ value any point can have because $s_k \in [0, 1]$. Since $w^*$ is largest when $d = 4$ and since $\varepsilon = 1/7$ we have the upper bound $w^* < 1/\sqrt{5}$, which will be used in the following calculations.

**Lemma D.3.** $\phi_R = \cos^{-1}\left(\frac{d+2+s_{min} s_{max}(w^*)^2}{\sqrt{d+3+(s_{min}w^*)^2}\sqrt{d+3+(s_{max}w^*)^2}}\right)$ *where $s_{min}$ and $s_{max}$ are, respectively, the minimum and maximum values in $S$.*

*Proof:* We consider the angle between two points in different clusters and two points in the same cluster. Let $p_j^i, p_k^i, p_l^m$ be three arbitrary points in $R$ where $j \neq k$ and $m \neq i$.
The inter-cluster angle is $\mathsf{angle}(p_j^i, p_l^m) = \cos^{-1}\left(\frac{d+2+w(p_j^i)w(p_l^m)(w^*)^2}{\sqrt{d+3+(w(p_j^i)w^*)^2}\sqrt{d+3+(w(p_l^m)w^*)^2}}\right)$.
The intra-cluster angle is $\mathsf{angle}(p_j^i, p_k^i) = \cos^{-1}\left(\frac{d+3+w(p_j^i)w(p_k^i)(w^*)^2}{\sqrt{d+3+(w(p_j^i)w^*)^2}\sqrt{d+3+(w(p_k^i)w^*)^2}}\right)$.

Note that the inter-cluster angle is maximized when $w(p_j^i) = s_{max}$ and $w(p_l^m) = s_{min}$. This angle is exactly what we claim $\phi_R$ to be. So it is left to show that this maximized inter-cluster angle is always larger than any intra-cluster angle. Note that any inter-cluster angle is lower bounded by $\mathsf{angle}(p_j^i, p_l^m) \geq \cos^{-1}\left(\frac{d+2+1/5}{d+3}\right)$ and thus we have

$$\mathsf{angle}(p_j^i, p_k^i) \leq \cos^{-1}\left(\frac{d+3}{d+3+\frac{1}{5}}\right) < \cos^{-1}\left(\frac{d+2+\frac{1}{5}}{d+3}\right) \leq \mathsf{angle}(p_j^i, p_l^m). \qquad \blacksquare$$

**Lemma D.4.** *The angular diameter of $P$, $\phi_P$, is equal to $\phi_R$.*

*Proof:* To prove the lemma we show the lower bound for $\phi_R$, seen in Lemma D.3, is larger than $\mathsf{angle}(p_j^i, q)$, $\mathsf{angle}(p_j^i, q')$, and $\mathsf{angle}(q, q')$ for a point $p_j^i \in R$. First note that by Definition D.2, $\mathsf{angle}(q, q') = \varepsilon \phi_R < \phi_P$, since $\phi_P$ is at least $\phi_R$.

It is clear that $\mathsf{angle}(p^i_j, q) \leq \mathsf{angle}(p^i_j, q')$, so we only need to consider the latter. We first find an upper bound on $\mathsf{angle}(p^i_j, q')$. Because $||q||^2 \leq (d + 2 + 1/d + 1/20)$, we obtain

$$\cos(\mathsf{angle}(p^i_j, q')) \geq \frac{(d+1)^2}{d\sqrt{d + 3 + \frac{1}{5}}\sqrt{d + 2 + \frac{1}{d} + \frac{(w^*)^2}{4} + ||q||^2 \tan^2(\varepsilon \phi_R)}}$$

$$\geq \frac{(d+1)^2}{d\sqrt{d + \frac{16}{5}}\sqrt{d + \frac{1}{d} + \frac{23}{10}}}.$$

The last inequality holds because $\phi_R \leq \cos^{-1}\left(\frac{d+2}{d+4}\right)$ and $||q||^2 \tan^2(\varepsilon \cos^{-1}\left(\frac{d+2}{d+4}\right)) < 1/4$ for all $d \geq 4$ and $\varepsilon = 1/7$. To verify $\mathsf{angle}(p^i_j, q') \leq \phi_R$, we use the this upper bound on $\mathsf{angle}(p^i_j, q')$ and the same lower bound on $\phi_R$ used in the proof of Lemma D.3. The limit of the ratio of these two bounds is

$$\lim_{d \to \infty} \frac{\cos^{-1}\left(\frac{d^2 + 2d + 1}{d\sqrt{d + 3 + \frac{1}{5}}\sqrt{d + \frac{1}{d} + \frac{23}{10}}}\right)}{\cos^{-1}\left(\frac{d + 2 + \frac{1}{5}}{d+3}\right)} = \sqrt{\frac{15}{16}} < 1.$$

Since the derivative of this ratio is positive for all $d \geq 4$, this limit is an upper bound. Thus, $\mathsf{angle}(p^i_j, q') \leq \phi_R$ and hence $\phi_P = \phi_R$. ∎

**Corollary D.5.** *The diameter of $P$ is less than $\pi/4$.*

*Proof:* From the above proofs we have the bound $\phi_P = \phi_R \leq \cos^{-1}\left(\frac{d+2}{d+3+\frac{1}{5}}\right)$. Thus for any $d \geq 4$, $\phi_P < \pi/4$, and moreover this upper bound is a decreasing function whose limit is 0 as $d$ goes to infinity. ∎

We now prove the completeness and soundness of the above reduction using the fact that $\phi_P = \phi_R$ from Lemma D.4.

**Lemma D.6 (Completeness).** *If there is a subset $\{s_{k_1}, s_{k_2}, \cdots, s_{k_d}\}$ of $d$ values (not necessarily distinct) such that $\sum_{i \in [d]} s_{k_i} = d/2$, then the above described instance of Problem D.1 is a true instance, i.e. there is a $d$ sized subset $X \subseteq P$ such that $d_{conic}(X, P) \leq \varepsilon \phi_P$.*

*Proof:* Let $x_i = (u + e_i, s_{k_i} \cdot w^*, 0)$, which by Definition D.2 is a point in $P$ and let $X = \{x_1, \ldots, x_d\}$. By Observation C.3, $\max_{p \in P} \mathsf{d}(p, \mathsf{Conic}(X)) \leq \varepsilon \phi_P$ implies $d_{conic}(X, P) \leq \varepsilon \phi_P$. We now show that $\max_{p \in P} \mathsf{angle}(p, \mathsf{Conic}(X)) \leq \varepsilon \phi_P$ which completes the lemma.

First observe that for any point $p^i_j$ in $P$, $x_i$ will have the same values as $p^i_j$ in all the $a_i$ coordinates. Consider the right triangle formed by the origin, $p^i_j$ with $w(p^i_j)$ set to 0, and $p^i_j$ with $w(p^i_j)$ set to $w^*$. Both $p^i_j$ and $x_i$ lie somewhere on the short leg since their $w$ values are at most $w^* = \tan(\varepsilon \cos^{-1}\left(\frac{d+2+1/5}{d+3}\right)) \leq \tan(\varepsilon \phi_P)$. Thus their angle is less than the angle opposite the short leg. Since $\tan^{-1}(\cdot)$ is monotonically increasing and the length of the longer leg is at least $\sqrt{d}$, we get

$$\mathsf{angle}(p^i_j, x_i) \leq \tan^{-1}\left(\frac{w^*}{\sqrt{d}}\right) \leq \tan^{-1}(\tan(\varepsilon \phi_P)) \leq \varepsilon \phi_P.$$

The only other points in $P$ are $q$ and $q'$ and by definition $\mathsf{angle}(q, q') = \varepsilon \phi_P$. Again, we claim $x = \frac{1}{d}\sum_i^d x_i \in \mathsf{Conic}(X)$ is the point $q$ which implies $\max_{p \in P} \mathsf{angle}(p, \mathsf{Conic}(X)) \leq \varepsilon \phi_P$. As $X$ contains exactly one point from each set $P^i$, and in each such set all points have $a_i = 2$ and all other $a_j = 1$, it holds that $x$ has $\frac{1+d}{d}$ for all the $a$ coordinates. All points in $X$ have $v = 0$ and so this holds for $x$ as well. Thus we only need to verify that $w(x) = w(q) = \frac{w^*}{2}$, for which we have $w(x) = \frac{1}{d}\sum_i w(x_i) = \frac{w^*}{d}\sum_i s_{k_i} = \frac{w^*}{2}$. ∎

**Lemma D.7.** *Let $P \subset \mathbb{R}^{d+2}$ be as defined above, and let $X \subseteq P$ be a subset of size d. Then if $d_{conic}(X, P) \leq \varepsilon\phi_P$, then for all i, X contains exactly one point from $P^i$.*

*Proof:* Assume that there is a set $P^t$ such that $X \cap P^t = \emptyset$. Consider an arbitrary point $p_j^t$ in $P^t$. We prove that $\mathsf{angle}(x, p_j^t) > \varepsilon\phi_P$ for any point $x$ in $\mathsf{Conic}(X)$, which contradicts the assumption that $d_{conic}(X, P) \leq \varepsilon\phi_P$.

We first find a point $x'$ that lower bounds $\mathsf{angle}(x, p_j^t)$ for all $x$ in $\mathsf{Conic}(X)$. Since every point in $P \setminus P^t$ has at most $\frac{1+d}{d}$ in the $a_t$ coordinate, any point $x$ in $\mathsf{Conic}(X)$ will have at most this value for $a_t$. Thus, we let $x'$ have the same value as $p_j^t$ in every coordinate except for $a_t$ which is $\frac{1+d}{d}$. Note that the angle between $x'$ and $p_j^t$ is minimized when they have the same $w$ values. Thus we let $w(p_j^t) = w(x') = w$ and their angle becomes

$$\mathsf{angle}(x, p_j^t) \geq \mathsf{angle}(x', p_j^t) = \cos^{-1}\left(\frac{2 + \frac{2}{d} + d - 1 + w^2}{\sqrt{(d + 3 + w^2)\left(\left(\frac{1+d}{d}\right)^2 + d - 1 + w^2\right)}}\right)$$

The derivative with respect to $w$ of the above inner ratio is positive for any $d \geq 4$, $w \geq 0$. Therefore, as $w^* < 1/\sqrt{5}$ is an upper bound on the $w$ value of any point, we let $w(p_j^t) = w(x') = 1/\sqrt{5}$ which maximizes the inner ratio and consequently minimizes the angle between the two points. Thus for any $x \in \mathsf{Conic}(X)$,

$$\mathsf{angle}(x, p_j^t) \geq \mathsf{angle}(x', p_j^t) \geq \cos^{-1}\left(\frac{2 + \frac{2}{d} + d - 1 + \frac{1}{5}}{\sqrt{(d + 3 + \frac{1}{5})\left(\left(\frac{1+d}{d}\right)^2 + d - 1 + \frac{1}{5}\right)}}\right)$$

Remember from the proof of Lemma D.4 that $\cos^{-1}\left(\frac{d+2}{d+4}\right)$ is an upper bound for $\phi_P$. Thus to show $\mathsf{angle}(x, p_j^t) > \varepsilon\phi_P$ it suffices to argue the ratio

$$\frac{\cos^{-1}\left(\frac{2 + \frac{2}{d} + d - 1 + \frac{1}{5}}{\sqrt{(d + 3 + \frac{1}{5})\left(\left(\frac{1+d}{d}\right)^2 + d - 1 + \frac{1}{5}\right)}}\right)}{\cos^{-1}\left(\frac{d+2}{d+4}\right)},$$

is $> \varepsilon = 1/7$ for any $d \geq 4$. Thus the claim follows as the derivative of this ratio is positive for all $d \geq 4$, and thus the minimum occurs at $d = 4$, which itself is greater than $1/7$. ∎

**Lemma D.8.** *If $d_{conic}(X, P) \leq \varepsilon\phi_P$, then $q \in \mathsf{Conic}(X)$ and moreover $q = \frac{1}{d}\sum_{x_i \in X} x_i$.*

*Proof:* By Lemma D.7, $X$ cannot contain the point $q'$, and since all points in $P$ other than $q'$ have $v$ coordinate equal to 0, $\mathsf{Conic}(X)$ is contained in the hyperplane $v = 0$. Note that the projection of $q'$ onto this hyperplane is the point $q$. As the angle between $q'$ and $q$ is exactly $\varepsilon\phi_P$ by definition, it then must be that $q \in \mathsf{Conic}(X)$.

For the second part of the lemma, consider the coordinate $a_i$ for any $i$. $q$ has this coordinate set to $\frac{1+d}{d}$. Again by Lemma D.7, $X$ contains exactly one point with this coordinate set to 2 and all other points in $X$ have this coordinate set to 1. Thus the convex combination for $q$ must take at least a $1/d$ fraction of this one point in order to achieve a total value of $\frac{1+d}{d}$ in coordinate $a_i$. On the other hand, the convex combination cannot take more than a $1/d$ fraction of this point, as by symmetry it would imply for some other coordinate $a_j$, the total value would be strictly less than $\frac{1+d}{d}$. Thus for all $i$, the convex combination for $q$ must take exactly a $1/d$ fraction of $x_i$ ∎

**Lemma D.9 (Soundness).** *Let $P$ be an instance of Problem 3.1 generated from a d-SUM instance $S$, as described in Definition D.2. If there is a subset $X \subseteq P$ of size $d$ such that $d_{conic}(X, P) \leq \varepsilon\phi_P$, then there is a choice of d values from S that sum to exactly $d/2$.*

*Proof:* From Lemma D.7 we know that $X$ consists of exactly one point from each cluster $P^i$. Thus for each $x_i \in X$, $w(x_i) = s_{k_i} \cdot w^*$ for some $s_{k_i} \in S$. By Lemma D.8, $q = \frac{1}{d} \sum_i x_i$, which implies $w(q) = \frac{1}{d} \sum_i w(x_i)$. By Definition D.2 $w(q) = \frac{w^*}{2}$, which implies:

$$\frac{w^*}{2} = \frac{1}{d} \sum_i w(x_i) = \frac{w^*}{d} \sum_i s_{k_i}.$$

Thus we have a set $\{s_{k_1}, \ldots, s_{k_d}\}$ of $d$ values from $S$ such that $\sum_i s_{k_i} = d/2$. ∎

Lemma D.6 and Lemma D.9 immediately imply the following.

**Theorem D.10.** *For point sets in $\mathbb{R}^{d+2}$, Problem D.1 is d-SUM-hard.*

# E  Additional Experimental Results

Figure E.1: Additional experimental results for feature selection.

# F  Proofs from Section 4

**Lemma F.1 (Restatement of Lemma 4.5).** *The diameter of $P$, $\Delta_P$, is equal to $\Delta^*$.*

*Proof:* Observe that there are only five different types of pairs of points in $P$ that can determine $\Delta_P$. Consider three points $p_j^i, p_k^i, p_l^m$ in $P$ where $j \neq k$ and $m \neq i$, i.e. $p_j^i$ and $p_k^i$ are in the same cluster and $p_l^m$ is in a different cluster. The five distances are: $\mathsf{d}(p_j^i, p_k^i)$, $\mathsf{d}(p_j^i, p_l^m)$, $\mathsf{d}(p_j^i, q')$, $\mathsf{d}(p_j^i, q')$, $\mathsf{d}(q, q')$. Consider $\mathsf{d}(p_j^i, p_l^m)$. The distance between these two points is $\sqrt{2 + (\varepsilon w(p_j^i) - \varepsilon w(p_l^m))^2}$. This distance is maximized when $w(p_j^i)$ and $w(p_l^m)$ are the maximum and minimum values in $S$, which is exactly $\Delta^*$. Thus we need to show every other distance is less than $\Delta^*$.

First we have $\mathsf{d}(p_j^i, p_k^i) \leq \sqrt{(\varepsilon w(p_j^i) - \varepsilon w(p_k^i))^2} \leq \varepsilon$ and $\mathsf{d}(q, q') = \varepsilon \Delta^* < \Delta^*$. Next, note that $(\varepsilon s_j - \varepsilon/2)^2 \leq 1/36$ and $(\varepsilon \Delta^*)^2 \leq 7/27$ since $\varepsilon \leq 1/3$. Thus,

$$\mathsf{d}(p_j^i, q) \leq \mathsf{d}(p_j^i, q') = \sqrt{(1 - 1/d)^2 + (d-1)/d^2 + (\varepsilon s_j - \varepsilon/2)^2 + (\varepsilon \Delta^*)^2} < \sqrt{2} \leq \Delta^*. \quad \blacksquare$$

**Lemma F.2 (Restatement of Lemma 4.7).** *Let $P \subset \mathbb{R}^{d+2}$ be as defined above, and let $X \subseteq P$ be a subset of size $d$. If $d_{convex}(X, P) \leq \varepsilon\Delta$, then for all $i$, $X$ contains exactly one point from $P^i$.*

*Proof:* Assume that there is a set $P^t$ such that $X \cap P^t = \emptyset$. Consider an arbitrary point $p_j^t$ in $P^t$. We prove that $d(x, p_j^t) > \varepsilon\Delta$ for any point $x$ in $\mathsf{Convex}(X)$, which is a contradiction as it implies $d_{convex}(X, P) \leq \varepsilon\Delta$. This will imply each set $P^i$ has non-empty intersection with $X$, and as the size of $X$ is the same as the number of $P^i$ sets, this in turn implies $X$ contains exactly one point from each $P^i$ set.

First observe the only points in $P \setminus P^t$ with non-zero $a_t$ coordinate are $q$ and $q'$, each of which has $a_t = 1/d$. As $x$ is a convex combination of points from $P \setminus P^t$ this implies that its $a_t$ coordinate is at most $1/d$. On the other hand, $p_j^t$ has coordinate $a_t$ set to 1. This implies that $d(x, p_j^t) \geq \sqrt{(1 - 1/d)^2} = 1 - 1/d \geq 1/2 > \varepsilon\Delta^* = \varepsilon\Delta$. (Other coordinates can be ignored as differing values in other coordinates can only increase $d(x, p_j^t)$.) $\blacksquare$

**Lemma F.3 (Restatement of Lemma 4.8).** *If $d_{convex}(X, P) \leq \varepsilon\Delta$, then $q \in \mathsf{Convex}(X)$ and moreover $q = \frac{1}{d}\sum_{x_i \in X} x_i$.*

*Proof:* By Lemma 4.7, observe that $X$ cannot contain the point $q'$. On the other hand, all points in $P$ other than $q'$ have $v$ coordinate equal to 0, and thus $\mathsf{Convex}(X)$ is contained in the hyperplane $v = 0$. At the same time, $q'$ has $v$ coordinate equal to $\varepsilon\Delta^*$, and thus has distance $\varepsilon\Delta$ to the hyperplane $v = 0$. Thus the projection of $q'$ onto the $v = 0$ hyperplane must be contained in $\mathsf{Convex}(X)$. However, the projection of $q'$ onto this hyperplane is the point $q$, and thus $q \in \mathsf{Convex}(X)$.

For second part of the lemma, consider the coordinate $a_i$ for any $i$. $q$ has this coordinate set to $1/d$. Again by Lemma 4.7, $X$ contains exactly one point with this coordinate set to 1 and all other points in $X$ have this coordinate set to zero. Thus any convex combination of $X$ realizing $q$ must take a $1/d$ fraction of this point. As this is true for all $i$, the claim follows. $\blacksquare$