[Reviews · NeurIPS 2017]

Reviewer 1



The paper "Sparse approximate conic hulls" develops conic analogues of approximation problems in convex geometry, hardness results for approximate convex and conic hulls, and considers these in the context of non-negative matrix factorization. The paper also presents numerical results comparing the approximate conic hull and convex hull algorithms, a modified approximate conic hull algorithm (obtained by first translating the data), and other existing algorithms for a feature-selection problem. Numerical results are also presented. The first theoretical contribution is a conic variant on the (constructive) approximate Caratheodory theorem devised for the convex setting. This is obtained by transforming the rays (from the conic problem) into a set of vectors by the "gnomic projection" applying the approximate Caratheodory theorem in the convex setting, and transforming back. The main effort is to ensure the error behavior can be controlled when the gnomic projection/its inverse are applied. This requires the points to have angle to the "center" strictly smaller than pi/2. This kind of condition on the points persists throughout the "conic" results in the paper. The second main contribution is establishing hardness results for approximate conic and convex hull problems. Finally, the main point of the paper seems to be that the epsilon-approximate conic hull algorithm can be used to approximately solve NMF under the column subset restriction. The paper is quite well written, albeit a little dense, with much of the core technical content relegated to the supplementary material. The results are interesting from a purely convex geometry point of view. Nevertheless, I am somewhat unconvinced about certain issues in the formulation of the problems in the paper (rather than the technical results of the paper, which are nice): -- the choice of angular metric seems a bit arbitrary (there are other ways to put a distance on the non-negative orthant that is naturally adapted to the conic setting, such as the Hilbert metric. Perhaps the angular metric is best suited to using Frobenius norm error in the matrix factorization problem? If so it would be great if the authors could make this more clear. -- The gamma-shifted conic version performs well in experiments (and the original conic version does not), which is interesting. How should we choose the shift though? What reason is there not to make the shift very large? Is it possible to "pull back" the gamma shift throughout the paper, and formulate a meaningful version of approximate Caratheodory that has a parameter gamma? Perhaps choosing the best-case over gamma is a more interesting formulation (from a practical point of view) than the vanilla conic version studied in this paper. In addition to these concerns, I'm not sure how much innovation there is over the existing convex method of Blum et al, and over the existing hardness results for NMF in Arora et al. (I am not expert enough to know for sure, but the paper feels a little incremental). Minor comments: -- p3 line 128: this is not the usual definition of "extreme" point/ray in convex geometry (although lines 134-136 in some sense deal with the difference between the usual definition and the definition in this paper, via the notion of "degenerate") -- p4 line 149: one could also consider using the "base" of the cone defined by intersection with any hyperplane defined by a vector in the interior of the dual cone of the points (not just the all ones vector) instead of using the gnomic projection. This preserves the extreme points, so the basic strategy might work, but perhaps doesn't play nicely with Frobenius norm error in the NMF problem? It would be helpful if the authors could briefly explain why this approach is not favorable, and why the gnomic projection approach makes the most sense. -- p5 line 216, theorem 3.4: It would be useful if the authors describe the dependence on gamma in the parameters of the theorem. It seems this will be very bad as gamma approaches pi/2, and it would be good to be upfront about this (since the authors do a good job of making clear that for the problem setup in this paper the bounded angle assumption is necessary) -- p6 line 235, there is some strange typesetting with the reference [16] in the Theorem statement. -- p6 line 239: again the dependence on gamma would be nice to have here (or in a comment afterwards). -- p6 line 253: it may be useful to add a sentence to say how this "non-standard" version of d-sum differs from the "standard" version, for the non-expert reader. -- p8 lines 347-348: this is an interesting observation, that the gamma-shifted conic case is a sort of interpolant between the convex and conic cases. It would be interesting to be able to automatically tune gamma for a given scenario.

Reviewer 2



This paper provides an approximation algorithm for NMF. Specifically, the algorithm outputs few columns of the data matrix such that the conic hull of those columns is close to the conic hull of the columns of the entire matrix (in an appropriately defined metric). The main difference from existing works is that it does explicitly not assume there is a true model (i.e., few columns such that other columns are generated as combinations of those columns) or the separability assumption made in the existing works. In this sense, the results are model-free. The algorithm is based on gnomic projections and is heavily based on [16], with appropriate modifications. It is not clear why the algorithm does not depend explicitly on the dimensionality of the matrices. It would be better if the authors explain why clearly or point out which assumption leads to this effect. The results provided are interesting and would be of interest to the community. Hence I propose to accept the paper.

Reviewer 3



The paper presents a greedy approximation algorithm for finding a small subset S of columns of X such that conic hull of S approximates the conic hull of X. If k is the smallest number of columns to get the e -approximation, the algorithm produces O(k/e^2/3) columns that give O(e^1/3) approximation. The algorithm is heavily inspired from an earlier work [16] that produces same approximation for convex hull problem. Authors transform conic hull problem to that of convex hull using gnomonic projection (scaling the points so that they lie on a suitable hyperplane at unit distance from the origin). The main contribution is claimed to be the analysis of this which authors say is not immediate from the analysis in [16]. Apart from this, the paper also proves an approximate Caratheodory theorem for conic hulls, and shows that both convex and conic hull versions are d-SUM-hard. Overall, the paper shows that same approximation results hold for conic case as shown earlier for the convex case in [16]. I have not gone through all the proofs in the appendix so cannot comment on the immediacy of the proofs for the conic case from the convex case [16]. Approx Caratheodory theorem (thm 3.4) for conic seems to be not so difficult to obtain from the convex case though, given the monotonicity of the distortion on the hyperplane as a function of distortion on the sphere. Here are my other comments: 1. The paper should also cite earlier work on approximate solution for the conic hull problem where the quality is measured using some other metrics, eg. "Robust Near-Separable Nonnegative Matrix Factorization Using Linear Optimization", Gillis and Luce, 2015). 2. The paper keeps switching b/w the case when X is nonnegative matrix and the case when X is allowed negative entries. For ex, lines 41-43 talk about the nonnegative X case, whereas Algorithm 1 seems to be taking about general X (P can have points anywhere in R^m). 3. Algorithm 1: for P anywhere in R^m, the algorithm takes q as any vector in the space. However I think q should be in the dual cone of conic(P) to make the algorithm work. For nonnegative data, the dual cone is positive orthant itself. 4. Lines 148-150: the statement about earlier work [11] is not right -- they also allow any vector q in the positive orthant. See Algorithm 1 in [11] "detection step" and "Remarks (3)" in the same paper. 5. What optimization problem is solved to project points onto the current convex hull at step k? I didn't see the paper talking about this. 6. How is mutant X-RAY (line 316) is related to Algorithm 1 in the paper? Essentially if denominator (p^T X_j) is removed in the "detection step" in [11] and "max" variant is used, this is "mutant X-RAY" as called in the paper. 7. Will Algorithm 1 correctly recover the set of columns S if the data is generated using conic combinations of X, i.e., for the case when X = conic(S)? It doesn't look like so. A formal proof or comment would be good.